# A Top-N Movie Recommendation Framework Based on Deep Neural Network with Heterogeneous Modeling

**Jibing Gong** [1,2,3,*,†] , **Xinghao Zhang** [1,2,*,†] , **Qing Li** [1,2] , **Cheng Wang** [1,2] , **Yaxi Song** [1,2] and **Zhiyong Zhao** [1,2] and **Shuli Wang** [2,3,4]

1   School of Information Science and Engineering, Yanshan University, Qinhuangdao 066004, China; liqing1@stumail.ysu.edu.cn (Q.L.); wangcheng_ysu@163.com (C.W.); 13293151871@163.com (Y.S.); ys_zhaozhiyong@163.com (Z.Z.)
2   The Key Lab for Computer Virtual Technology and System Integration of Hebei Province, Yanshan University, Qinhuangdao 066004, China; wangshuli@ysu.edu.cn
3   Key Laboratory for Software Engineering of Hebei Province, Yanshan University, Qinhuangdao 066004, China
4   School of Science, Yanshan University, Qinhuangdao 066004, China
*   Correspondence: gongjibing@ysu.edu.cn (J.G.); 201922040145@stumail.ysu.edu.cn (X.Z.)
†   These authors contributed equally to this work.

**Abstract:** To provide more accurate and stable recommendations, it is necessary to combine display information with implicit information and to dig out potential information. Existing methods only consider explicit feedback information or implicit feedback information unilaterally and ignore the potential information of explicit feedback information and implicit feedback information, which is also crucial to the accuracy of the recommendation system. However, the traditional Heterogeneous Information Networks (HIN) recommendation ignores the attribute information in the meta-path and the interaction between the user and the item and, instead, only considers the linear characteristics of the user-object often ignoring its non-linear characteristics. Aiming at the potential information acquisition problem from assorted feedback, we propose a new top-N recommendation method MFDNN for Heterogeneous Information Networks (HINs). First, we consider explicit and implicit feedback information to determine the potential preferences of users and the potential features of the product. Then, matrix factorization (MF) and a deep neural network (DNN) are fused to learn independent feature embeddings through MF and DNN, and fully considering the linear and non-linear characteristics of the user-object. MFDNN was tested on several real data sets, such as Movie-Lens, and compared with benchmark experiments. MFDNN significantly improved the hit ratio (HR) and normalized discounted cumulative gain (NDCG). Further research showed that the meta-path bias had an excellent effect on the gain of potential information mining and the fusion of explicit and implicit information in the accuracy and stability of user interest classification.

**Keywords:** deep neural network; matrix factorization; top-N recommendation; implicit feedback information; meta-path bias

## 1. Introduction

A Recommender System (RS), a program that attempts to recommend the most suitable products/services to a user, aims at providing personalized services by retrieving the most relevant information and services from the big data generated on open, private, social, and IoT (Internet of Things) data islands [1]. With the rapid increase in the amount of information, when many users are looking for information about learning [2], movies [3], music [4], popular events [5], and other fields, how to quickly and accurately obtain the information they need most has become a key problem that needs to be solved in the current development of big data. The emergence of the recommendation system provides an opportunity to alleviate this problem [6].

With the development of data-mining algorithms, recommendation systems are used in information retrieval (e.g., Google and Baidu), news feeds (e.g., Toutiao and Google

News), e-commerce [7] (e.g., Amazon, Taobao, and Alibaba), and social networks (e.g., Facebook, Tencent, and Twitter) have achieved great success in various fields, effectively alleviating the contradiction between information and users. Recommendation systems due to their multi-domain applicability are among the main topics of scientific interest in recent years [8].

Today, almost every organization leverages Recommender Systems to better understand their customers and to suggest products and services [1]. For example, in the field of e-commerce, recommendation systems are used to personalize products recommended to users; and, in the field of short videos, recommendation systems are designed to personalize and recommend short videos that users will love [9].

The recommendation system mines user interests from big data, captures interest changes in real time [10], quickly feeds back user needs, helps customers complete data access work with simple operation procedures and comprehensive data analysis [11], and formulates or adjusts user recommendation information in a targeted manner. The security and user experience the efficiency of production and life are greatly improved, the process of information interaction, commodity circulation, and industrial asset circulation are accelerated, and social development and the improvement of people's living standards are effectively promoted.The research by Ricci et al. [9] showed that recommendation systems have been ubiquitous in various fields, including movies, music, travel, video, news, books, and general products.

There are three main combination strategies for hybrid recommendation [12]: pre-fusion, middle fusion, and post-fusion. (1) Pre-fusion refers to the fusion of multiple recommendation algorithms in the process of constructing a recommendation model, combining them into a unified model, performing a feature extraction training model, and then generating recommendation results based on the fusion model. (2) Middle fusion is based on one recommendation algorithm as the framework while fusing another recommendation algorithm. (3) Post-fusion means that each recommendation algorithm is trained separately to generate recommendation results, and finally a combination strategy is adopted to fuse the recommendation results of each recommendation model. Combination strategies that can be adopted include simple voting, linear combination, etc.

Recommendation systems rely on user feedback to evaluate attitudes toward items viewed by users. According to the nature of user feedback [6], this can be divided into explicit user feedback (for example, ratings, likes, and dislikes) or implicit feedback (for example, clicks, plays, and views), that is, display feedback and implicit feedback [7]. Explicit feedback is that the user's preferences can be directly expressed and exist in a way that makes it easy to obtain their preferences. Implicit feedback refers to the user's preference behavior information expressed in an indirect way rather than directly [13].

Explicit feedback data has the ability to express user preferences and behaviors more accurately; however, in real life, it is difficult to obtain representative and sufficient amount of explicit feedback information based on users [14]. At the extreme, feedback data can be very scarce and not easy to obtain in many application scenarios. Implicit information data is easy to obtain, and the amount of information is relatively large; however, its information cannot accurately express user preferences. Therefore, if we make full use of the advantages of the two types of data, we can achieve a good recommendation effect [8].

For the first time, ROber and Yen [15] proposed the theoretical idea of combining explicit feedback data with implicit feedback data. Nathan N. Liu et al. [16] proposed a matrix factorization model, which used different weights for implicit feedback data and explicit feedback data for learning modeling. The main idea of the matrix factorization model is to treat purchased and viewed commodities as implicit feedback data and mark them as "1", to mark other types of commodities as "0" for processing, and then combine them with display feedback data, before combining explicit feedback and implicit feedback.

Weike [17] first clustered user sets and item sets to propose a learning model. GaiLim [18] proposed a personalized ranking model that combined explicit and implicit feedback, which was implemented by optimizing the evaluation index ERR (Expected Reciprocal Rank).

With the rapid development of deep learning, the application of deep learning in recommendation models is gradually increasing. Ding et al. [19] proposed a friend recommendation model based on a Bayesian ranking deep neural network, which converted the recommendation problem into a ranking problem. The recommendation method based on deep learning has been successfully applied to label recommendation [20] and POI [21] recommendation, and different neural network structures have been proposed, such as multi-layer perceptron (MLP), convolutional neural networks (CNN), and recurrent neural networks (RNN) [6].

A deep neural network can effectively simulate nonlinearity in the data through nonlinear activation. Some are also used to transform recommendation problems into classification problems. Although deep learning has been widely used in recommendation methods and recommendation systems, the research on recommendation methods based on deep learning is still in the development stage [10].

Recently, some researchers have realized the importance of heterogeneous information for recommendation. Heterogeneous Information Networks (HINs) effectively integrate more information and form a new trend in the development of data mining. A large amount of user information can be obtained to make the content of the recommendation system more diverse, including academics, commodities, friends, music, services, etc. [22]. In addition to traditional recommendation methods, a large number of new recommendation methods have also been generated, such as social network-based recommendation methods, context awareness recommended methods of the internet [23], and location-based recommendations [24].

However, these recommendation methods based on heterogeneous information also have some challenges: (1) Massive amounts of HIN data hide the objects' comprehensive and detailed information. Hence, mining and analyzing valuable information for HIN recommendations is a key challenge. (2) The rapid expansion of HINs generates increasing amounts of data, such as a wide variety of user features. How to take advantage of these features to build a unified top-N recommendation model is a substantial problem. (3) It is difficult to combine and measure all of the features of objects to produce HIN recommendations. Considering all of the features may require a significant amount of time and cause an over-fitting problem. (4) Transmission delays, energy saving issues, data redundancy, and inaccuracy of data transmission during data transmission are also issues that need to be resolved [25]. As a result, selecting the most relevant one from the recommendation results among all the features of objects in HINs is challenging.

Based on user–item history information, rating prediction models predict the specific rating for an item given by a user [26]. In practical applications, merchants are concerned about whether users will buy an item, which means telling whether a user will watch a movie is more consequential than predicting the rating value that the user may give after watching the movie.

This study mainly considers a bipartite network, a special type of heterogeneous information network, for generating top-N recommendations [27]. Existing user–item data recommendation methods mostly consider user–item implicit feedback and ignore the user preference characteristics behind the explicit data. Here, the explicit feedback data refers to the user–item rating information, and the implicit feedback refers to whether user–item interaction information exists or not.

To obtain the user's preference information more comprehensively, this study considers both the explicit and implicit feedback from user–item interactions for mining the users' potential preferences and the underlying features of items [28]. In order to improve the performance of the recommendation algorithm of the heterogeneous information network, the recommendation model is constructed by fusing matrix factorization (MF) [29] and a deep neural network (DNN) [19], which, respectively, obtain the explicit and implicit feedback prediction results.

The explicit and implicit feedback prediction results are combined to generate the top-N recommendations [30–32]. Two bias factors are introduced to consider the characteristics

of the user–item data in explicit feedback information. More specifically, we first consider both explicit and implicit feedback data; the explicit and implicit feedback information is separately trained as input to better mine the potential information behind the user–item rating meta-path information. Here, explicit feedback data refers to user–item ratings, and includes both meta-path attribute information and the characteristics of objects; implicit feedback refers to the user–item relationship data.

Then, the MF and DNN models are merged to form the relationship between the user–item rating meta-path and the attribute value information. As they learn embedded features independently, MF and DNN can fully consider the linear and nonlinear user–item features, and respectively train the explicit and implicit feedback data in order to obtain the corresponding output. Subsequently, by combining the explicit and implicit feedback prediction results, the top-N items are recommended to the target user.

Finally, we use the MovieLens dataset to verify the model and apply leave-one-out to further evaluate the model. The performance of the method is evaluated using the hit ratio (HR) and normalized discounted cumulative gain (NDCG) evaluation metrics. The proposed method was found to outperform the traditional recommendation model and state-of-the-art recommendation methods.

The contributions of this paper are summarized as follows.

- We exploit both explicit and implicit feedback information to obtain the user's preference information and the underlying characteristics of the item based on the meta-path selection results. Additionally, in order to obtain explicit feedback information, two bias factors are introduced according to the individual characteristics of the user–item information.
- We fuse MF and DNN to mine the potential features of users and items from both linear and nonlinear perspectives. MF and DNN learning are independently embedded to better capture user preference information and the potential feature information of items.
- Using the leave-one-out evaluation method, we combine explicit and implicit feedback results to obtain the top-N recommendation list for target users and adopt the HR and NDCG metrics to evaluate the proposed model.

The remainder of this paper is organized as follows: We briefly outline the related work in Section 2. We provide the problem definition and explain the proposed architecture in Section 3. Section 4 shows and discusses the experimental results that validate our model. Finally, we conclude this paper in Section 5.

The notations used in this paper are summarized in Table 1:

**Table 1.** Notations

| Symbol | Description |
| --- | --- |
| $Y$ | User–item interaction matrix |
| $U$ | Set of users |
| $I$ | Set of items |
| $\hat{Y}$ | Final prediction results |
| $\hat{Y}^-$ | Implicit feedback prediction results |
| $\hat{Y}^+$ | Explicit feedback prediction results |
| $Y^-$ | User–item relation matrix |
| $Y^+$ | User–item rating matrix |
| $\hat{Y}_{ui,MF}$ | Final prediction results of MF |
| $\hat{Y}_{ui,DNN}$ | Final prediction results of DNN |
| $p_u^F$ | User embedding vector of MF |
| $q_i^F$ | Item embedding vector of MF |
| $p_u^I$ | User embedding vector of DNN |
| $q_i^I$ | Item embedding vector of DNN |

## 2. Related Work

An increasing number of researchers have focused on HIN recommendations with different types of objects or relations [33]. Since HINs were first proposed in [34], many HIN recommendation methods have been proposed. In these works, similarity measurements are vitally important and fundamental, and the most popular method is path-based. For example, a meta-path associated with top-N similarity measurement was proposed in [34,35] proposed a recommendation method based on personalized semantics to predict users' ratings of items, and [36] proposed symmetric measurements on arbitrary meta-paths. Random-walk-based methods are usually used to mine the paths, weigh the paths, and compute the closeness or relevance between two nodes in a HIN [37].

Random walks in the connected components of the graph assume the properties of Markov Chains (steady-state distribution, irreducibility, etc.) [38,39]. However, these traditional HIN techniques ignore the value of link attributes; as a result, the meta-path cannot accurately capture the relationship between objects [40]. More recently, other strategies have been proposed to alleviate this shortcoming. A unified and flexible personalized sorting framework, MFPR, was proposed in [41]; this framework combines explicit feedback with multiple implicit feedback. In [42], a unified model fusing generalized matrix factorization and multilayer perception was proposed. In [43], a collaborative filtering recommendation method in view of heterogeneous relations was proposed.

Among the works described above, we must compare MFDNN against the model proposed in [44], because it is not only a state-of-the-art HIN recommendation method but also very similar to our model. DeepMF [44] performs click-through rate (CTR) prediction by combining the recommendation ability of factorization machines with the feature learning ability of deep learning, and simultaneously learns the low-order and high-order feature interactions from the original features of the input.

Compared with DeepMF, MFDNN has three main differences: (1) In MFDNN, MF and DNN learn embedded features separately, while DeepMF shares the same raw input feature vector. (2) The input layer of MFDNN combines user–item explicit and implicit feedback, while the input layer of DeepMF is a one-shot encoding of each feature field (e.g., gender, and location). (3) The output of MFDNN is a top-N recommendation list, while DeepMF aims to predict the value of ratings.

There are many other HIN recommendation algorithms, including collaborative filtering [45] and content-based recommendation [46]. These traditional methods were extensively used in the early phases of HIN. However, the former is not applicable to high-dimensional data, and there is a cold-start problem [47], while the latter takes fewer attributes into consideration.

Deep neural networks (DNN) have demonstrated breakthroughs in data mining, e.g., voice recognition [48], image labeling [49,50], and text classification [51–53]. Deep learning-based methods, which can learn a large-scale nonlinear network structure and obtain deep feature representations of users and items, have proven effective in recommendation tasks [54–56]. Convolutional neural networks have a powerful ability to learn feature representations and have the potential to learn sophisticated feature interactions [57,58].

BayDNN was proposed in [19] as a Bayesian personalized ranking deep neural network model for social network friend recommendations; in this model, the recommendation problem is regarded as a ranking problem. The method described in [59] adopts a simple pre-training strategy using a four-layer neural network for link prediction. In [21], a deep content-aware point-of-interest (POI) recommendation (DCPR) algorithm was proposed; broad learning from multiple sources of information is utilized to solve the problem. Based on the above studies, we found that deep learning-based recommendation methods are still in their infancy, and MFDNN effectively improves the accuracy of HIN recommendation.

In the past few decades, numerous researchers have focused on designing and implementing top-N recommendation methods; however, these methods only consider the direct relations between pairs of items to compute the similarities needed for constructing recommendation frameworks. In fact, a high-order information and neighborhood-based

method was proposed to merge high-order information earlier in the process; however, it did not significantly improve performance.

The sparse linear method (SLIM) was proposed in [60]; this method aggregates users' purchase/rating profiles to generate recommendation results. However, it can only model the relationship between items co-purchased by at least one user. To address the limitations of SLIM, LorSLIM [61], which introduces a low-rank structure, was proposed. Low-rank assumptions are usually driven by factor models. HOSLIM was proposed in [62], which revisited the problem of using higher-order information rather than low-rank information.

### 3. Our Approach: MFDNN

*3.1. Problem Definition*

In this subsection, we first provide the related preliminary definition and then provide a formal problem definition.

**Definition 1.** *Heterogeneous information network (HIN). HINs were first defined in [34]. A directed graph $G = \langle V, E \rangle$ is defined to present an information network, where V is the set of objects, E is the set of relations, the object-type mapping function is $\phi : V \to A$, and the relation-type mapping function is $\psi : E \to R$. A network is called a heterogeneous information network when the types of objects $|A| > 1$ or the types of relations $|R > 1|$; otherwise, it is a homogeneous information network.*

*A bibliographic information network [63] is a typical HIN that contains three types of objects: author, venue, and paper, and two types of relations: publish and write. Other examples of HINs are shown in [64]. We mainly focus on the bipartite network—a special HIN that has two types of objects.*

**Definition 2.** *Bipartite Network. A bipartite network is a special HIN that has two types of objects.*

**Problem 1.** *$U = \{u_1, u_2, \cdots, u_m\}$ is a user set of size m, and $I = \{i_1, i_2, \cdots, i_n\}$ is an item set of size n. We first analyze and select a reasonable meta-path that can help find the most similar user or item according to the meta-path. An example of a selected meta-path is shown in Figure 1.*

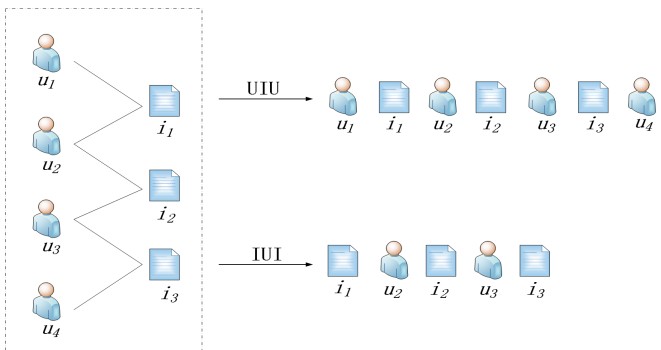

**Figure 1.** The selected meta-path.

In this study, we consider the meta-path link information as well as the attribute information of the user and item. According to the *UI* rating meta-path, we define the user–item interaction matrix $Y^- \in R^{m \times n}$ and the user–item rating matrix $Y^+ \in R^{m \times n}$ according to the historical rating record as Equation (1) and (2):

$$y_{ui}^- = \begin{cases} 1 & \text{if interaction (user } u, \text{ item } i) \text{ is observed(score>2);} \\ 0 & \text{otherwise.} \end{cases} \tag{1}$$

$$y_{ui}^+ = \begin{cases} Y_{ui}^+ & \text{the rating (user } u, \text{ item } i); \\ 0 & \text{otherwise.} \end{cases} \tag{2}$$

Here, a value of 1 for $y_{ud}^+$ denotes that $u$ and $i$ have an interaction; however, this does not necessarily mean that $u$ actually likes $i$. Similarly, a value of 0 for $y_{ud}^-$ also does not indicate that user $u$ dislikes item $i$; perhaps user $u$ is not aware of item $i$ at all. In other words, observed relations reflect the users' preferences on items, while unobserved relations can result from missing data. For example, when shopping online, the rating value of an item is affected by factors other than the item itself, such as delivery speed and service attitude.

In such cases, the final rating may not indicate whether the user likes the item. However, if the user buys the item, it is certain that one of the characteristics of the item attracts the user. Therefore, we simply conclude that the level of the rating reflects the preferences of the user. It becomes a challenge to learn users' intentions from the historical rating record since it contains various noisy data indicating users' preferences. We often cannot obtain explicit feedback information directly, and the data is sparse.

In contrast, we can easily obtain implicit feedback information, and the data covers most users and objects; thus, it can mitigate the problem of sparse data to some extent [65–67]. We obtain a top-N recommendation list via modeling with a recommendation algorithm according to historic explicit and implicit feedback information.

### 3.2. Top-N Recommendation Architecture

In general, a user's preferences will not change significantly over a relatively short period of time. Therefore, our goal is to combine the explicit and implicit feedback information by fusing MF and DNN to predict the missing user–item interaction rating value $\hat{y}_{ui}$ and sort the rating values to obtain the top-N recommendation list. The main framework is shown in Figure 2.

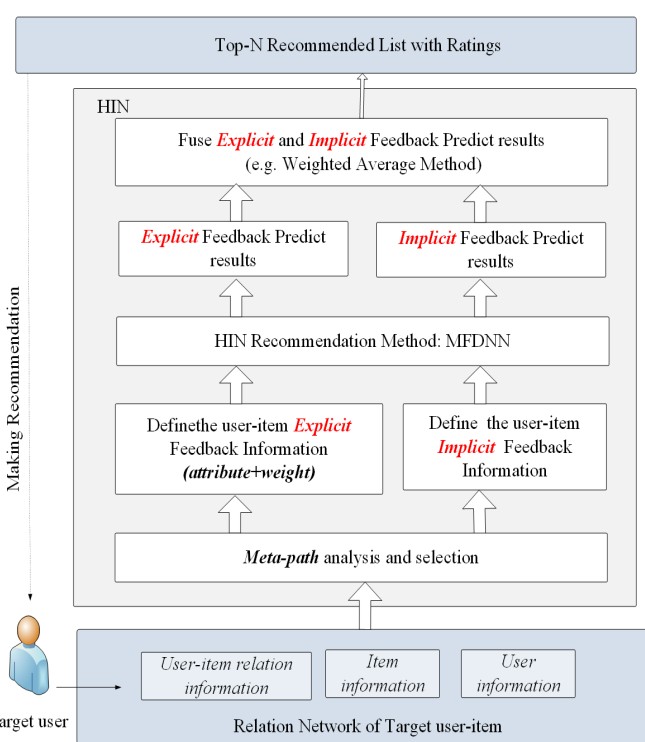

**Figure 2.** Architecture of the proposed method.

As shown in Figure 2, we first construct a user–item rating matrix and a user–item relation matrix, here, for the explicit information matrix construction, that is, the user–item rating matrix, we fill in the corresponding position according to the historical scoring record. For those without scoring, we fill in with 0; For the construction of the implicit information matrix, that is, the user–item relation matrix, according to the historical scoring record, we fill in the position corresponding to the user's score greater than 2 with 1, and

the other positions with 0; the user–item relation matrix and user–item rating matrix are made in different ways.

Then, we run the MFDNN model to obtain the explicit and implicit feedback prediction results; finally, we combine the explicit and implicit results to obtain the top-N recommendation list for target users. Here, we can choose the weighted average method (WAM) or simply sum the explicit and implicit prediction results. For explicit feedback prediction, we choose the parameters by minimizing the value of the cross-entropy loss between $y_{ui}^+$ and $\hat{y}_{ui}^+$, which is expressed by the formula in Equation (3):

$$L^+ = \sum_{(u,i) \in Y} log(\hat{y}_{ui}^+ + b_u + b_i) - \sum_{(u,i) \in Y^-} log(1 - \hat{y}_{ui}^+ - b_u - b_i). \tag{3}$$

This is the same as implicit feedback—the only difference is that explicit feedback information considers two individual bias factors $b_u$ and $b_i$, where $\hat{y}_{ui}^+$ denotes the explicit prediction results. By minimizing Equation (3), we can obtain the best recommendation list according to the explicit feedback. Additionally, we can obtain the results of $b_u$ and $b_i$ according to Equation (4) and (5):

$$b_u = \frac{\hat{y}_{ui}^+ - \bar{r}_u}{\#i}, \tag{4}$$

$$b_i = \frac{\hat{y}_{ui}^+ - \bar{r}_i}{\#u}. \tag{5}$$

where $\hat{y}_{ui}^+$ is the rating value that user $u$ has given to item $i$, $\bar{r}_u$ is the average of the ratings given by user $u$, and $\#u$ is the number of items that $u$ has rated. Similarly, $\bar{r}_i$ is the average rating of item $i$, and $\#d$ is the number of users that have rated $i$. By adding a regularization term to optimize the target loss function, the target loss function of the regular term is introduced as Equation (6):

$$L^+ = \sum_{(u,i) \in Y} log(\hat{y}_{ui}^+ + b_u + b_i) - \sum_{(u,d) \in Y^-} log(1 - \hat{y}_{ui}^+ - b_u - b_i) + \frac{\lambda}{2}(\|\hat{y}_{ui}^+\|^2 + b_u^2 + b_i^2). \tag{6}$$

where $\lambda$ is the regularization parameter.

### 3.3. Framework of MFDNN

In this section, we describe the design of MFDNN, a recommendation architecture based on MF and DNN. MF can fully consider the linear relation between users and items, on the other hand, DNN can fully consider the nonlinear features between users and items. The framework of MFDNN is shown in Figure 3.

The input data includes user–item explicit and implicit feedback, as shown in Figure 3. The explicit feedback is the user–item rating matrix constructed based on the meta-path, and the implicit feedback implies a user–item relation matrix. They are trained in order to obtain the corresponding results: $\hat{Y}^-$ and $\hat{Y}^+$. The embedding layers are independently trained for MF and DNN. For MF, the user is embedded as $p_u^F$, and the item is embedded as $q_i^F$, For DNN, the user is embedded as $p_u^I$, and the item is embedded as $q_i^I$.

The subsequent user and item embedding can be viewed as a potential vector for describing users and items in the context of a latent factor model. The embedding layer is a fully connected layer that maps the coefficient representation of the input layer to a

dense vector. The MF model and DNN model separately train the result and, finally, fuse the results by an activation function. This is shown as Equation (7):

$$\hat{y}_{ui} = \sigma(\hat{y}_{ui,MF} + \hat{y}_{ui,DNN}). \tag{7}$$

Here, we select the sigmoid function as the activation function because of the probability of $\hat{y}_{ui} \in [0,1]$. We note that $\hat{y}_{ui,MF}$ and $\hat{y}_{ui,DNN}$ are trained independently in the model.

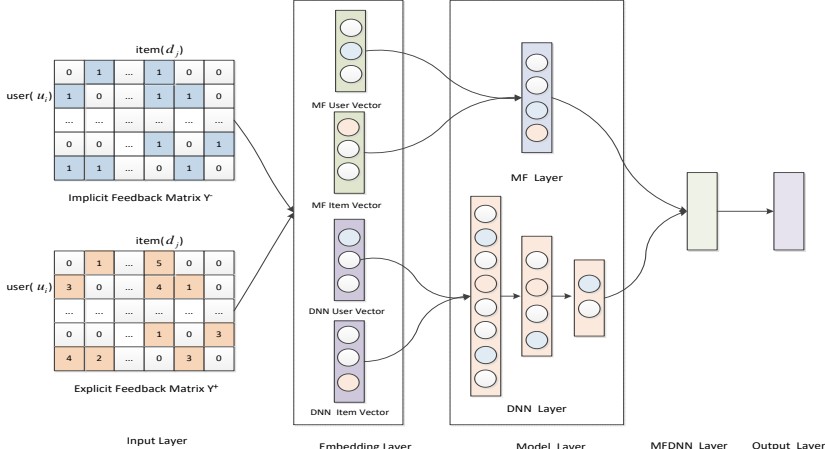

**Figure 3.** Framework of **MFDNN**.

### 3.4. Implementation of MFDNN

A linear combination of potential features of a user and an item can be learned by matrix factorization, and $p_u$ and $q_i$ are used to represent the potential vectors of $u$ and $i$, respectively. The matrix factorization estimates the inner product of $p_u$ and $q_i$ as the prediction function value, as shown in Equation (8):

$$\hat{y}_{ui} = p_u^T q_i = \sum_{k=1}^{K} p_{uk} q_{ik}. \tag{8}$$

where $K$ denotes the dimensions of latent space. However, using a simple inner product to estimate complex user–item interactions in the low-dimensional latent space limits the expression of MF and affects the generalization ability of the model. Thus, we define the mapping function of the first layer of MF as Equation (9):

$$\phi(p_u^F, q_i^F) = p_u^F \odot q_i^F. \tag{9}$$

where $\odot$ denotes the element-wise product of vectors; the output of MF is given by Equation (10):

$$\hat{y}_{ui,MF} = \alpha_{out}(h^T \phi(p_u^F, q_i^F)). \tag{10}$$

where $\alpha_{out}$ is an activation function. In consideration of convergence speed, we used the ReLU (rectified linear unit) function as the activation function, which is simply defined as max(0, x); $h^T$ is the weight vector. For DNN, we first obtain the first layer results by processing the embedding layer using Equation (11):

$$f_1 = \phi_1(p_u^I, q_i^I). \tag{11}$$

In the same manner, the second layer results are obtained using Equation (12):

$$f_2 = \alpha_2(W_2^T f_1 + b_2). \tag{12}$$

where $W_2^T$ and $b_2$ are the weight matrix and biased vector, respectively, and $\alpha_2$ is the activation function. The results of the N-th layer are obtained using Equation (13):

$$f_N = \alpha_N(W_N^T f_{N-1} + b_N).$$  (13)

According to Equations (11)–(13), we obtain the final DNN prediction result using Equation (14):

$$\hat{y}_{ui,DNN} = \alpha(W^{|H|+1} \cdot \left[p_u^I, q_i^I\right]^T + b^{|H|+1}).$$  (14)

where $\alpha$ is the activation function, $H$ is the number of hidden layers, and $W$ and $b$ are the weight matrix and biased vectors, respectively. Here, we chose the ReLU function as the activation function. The sigmoid activation function maps the output of each neuron to the (0,1) interval. This may hamper the performance of the model, and it is likely to cause an over-fitting problem; that is, when the output approaches 0 or 1, the neuron stops learning.

Although Tanh mitigates the problem of the sigmoid to a certain extent, the result is a scaled version of the sigmoid function. Therefore, the ReLU activation function was selected for the model. The ReLU activation function avoids over-fitting and supports sparse data so that the model does not overfit. The explicit and implicit prediction results are generated by MFDNN. After obtaining the $\hat{Y}^+$ and $\hat{Y}^-$ by executing MFDNN, we can obtain the final user–item prediction results according to Equation (15):

$$\hat{Y} = \omega_1 \hat{Y}^- + \omega_2 \hat{Y}^+ (0 \leqslant \omega_1 \leqslant 1, 0 \leqslant \omega_2 \leqslant 1).$$  (15)

where $\omega_1 + \omega_2 = 1$, $\omega_1$ is the weight of implicit feedback and $\omega_2$ is the weight of explicit feedback. As for the best recommendation list for target users, we find the optimal weights by minimizing the objective function using Equation (16):

$$\begin{aligned}
L &= \sum_{(u,i)\in Y} log\hat{y}_{ui} - \sum_{(u,i)\in Y^-} log(1 - \hat{y}_{ui}) + \frac{\lambda}{2}\left\|\omega_1^2 + \omega_2^2\right\| \\
&= -\sum_{(u,i)\in Y\cup Y^-} y_{ui}log\hat{y}_{ui} + (1 - y_{ui})log(1 - \hat{y}_{ui}) + \\
&\quad \frac{\lambda}{2}\left\|\omega_1^2 + \omega_2^2\right\|.
\end{aligned}$$  (16)

In the network structure, each layer employs fewer neurons in succession. By using a small number of hidden units at the upper layer, more abstract features can be learned from the data. For higher layers, the scale is reduced compared with the previous layer.

In addition, we utilized the dropout technique to alleviate the over-fitting problem. We chose the Adam algorithm [68] to train the model from scratch; this yielded faster convergence than SGD, which was important because we were unable to pay more attention to tuning the learning rate. The main steps of MFDNN are shown in Table 2:

**Table 2.** The MFDNN algorithm.

| Algorithm MFDNN algorithm |
| --- |
| $\tilde{Y}^- \Leftarrow$ User–item relation matrix; |
| $\tilde{Y}^+ \Leftarrow$ User–item rating matrix; |
| $\lambda \Leftarrow$ Parameter of regularization term; |
| Learning rate$\Leftarrow$0.001; |
| epochs $\Leftarrow$ Number of iterations; |
| $p_u^F \Leftarrow$ User embedding vector of MF; |
| $q_i^F \Leftarrow$ Item embedding vector of MF; |
| $p_u^I \Leftarrow$ User embedding vector of DNN; |
| $q_i^I \Leftarrow$ Item embedding vector of DNN; |
| epochs Calculate $\hat{y}_{ui,MF}^+$ Equations (8)–(10) |
| Calculate $\hat{y}_{ui,DNN}^+$ Equations (11)–(14) |
| Update MFDNN with Adam |
| Calculate $\hat{y}_{ui}^+$ Equation (7) |
| Calculate $\hat{y}_{ui}^-$ at the same way |
| Calculate $\hat{Y}$ Equation (15) |
| Top-N recommendation list |

## 4. Experiments

### 4.1. Experimental Setup

#### 4.1.1. Datasets

In this study, we used MovieLens 1m and Netflix, two benchmark datasets commonly used for testing recommendation systems, to evaluate the proposed model. These datasets do not require additional processing. We obtained the last interaction for every user, and then randomly selected 100 movies that the user had not interacted with. Table 3 shows the statistics for these two datasets.

**Table 3.** Descriptions of the MovieLens 1m and Netflix datasets

| Aspect | MovieLens 1m | Netflix |
| --- | --- | --- |
| #users | 6040 | 48,018 |
| #movies | 3706 | 17,770 |
| #ratings | 1,000,209 | 11,160,900 |
| Rating Density | 0.04468 | 0.01308 |

According to the definition of HIN and the content of [35,69], MovieLens 1m and Netflix are typical examples of HINs; the network pattern of these datasets are shown in Figure 4. Figure 4 shows that there are four types of objects: users, movies, actors, and directors. In this study, we only considered users and movies, which have the relations "rating" and "rated by." If two users often view the same movies, we can simply conclude that they have similar interests or preferences. Using the implicit and explicit feedback based on user–movie relations, we evaluated the performance of the proposed model.

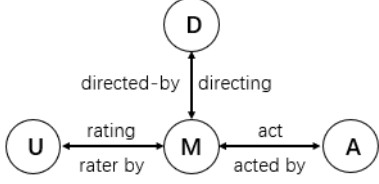

A:Actor,M:Movie,D:Director,U:User

**Figure 4.** The network pattern of MovieLens 1m and Netflix.

4.1.2. Evaluation Metrics

We adopted the HR (hit ratio) and NDCG (normalized discounted cumulative gain) metrics to evaluate the performance of MFDNN. HR: The probability that the user clicks or browses the recommended item. NDCG: Measures the quality of the ranking, which considers the ranking of the ratings; it is defined as

$$NDCG = \frac{DCG}{IDCG}. \tag{17}$$

The NDCG value of the first $k$ ratings is defined as

$$NDCG = \frac{DCG_k}{IDCG_k}. \tag{18}$$

where DCG is the discount cumulative gain. The DCG value of the first $k$ ratings is defined as:

$$DCG_k = \sum_{i=1}^{k} \frac{2^{rel_i} - 1}{log_2(i+1))}. \tag{19}$$

where $rel_i$ denotes the $i$th rating. IDCG denotes the ideal DCG, that is, the recommendation list sorted according to the value of ratings from high to low.

4.1.3. Baseline Methods

In this section, we aim to explain how our proposed MFDNN outperformed the existing top-N recommendation methods. We compare the MFDNN with the following representative methods in addition to two state-of-the-art recommendation methods (DMF [29] and NCF [42]) and three HIN-based methods (HeteCF [43], HeteMF [70], and CMF [71]).

**DMF (Deep Matrix Factorization)**: A new matrix decomposition model based on a neural network structure. It uses the user–item explicit feedback matrix as input and learns a common low-dimensional space of objects via a deep learning framework.

**NCF (Neural Collaborative Filtering)**: NCF can be used to express and generalize matrix decomposition under its framework. In order to use nonlinear enhanced NCF modeling, a multilayer perceptron is used to learn user–item interactions. NCF learning emphasizes the probability model of the binary properties of implicit data. It unifies the linear modeling advantages of MF and the nonlinear advantages of MLP to model the potential structure of user-projects.

**HeteCF (Heterogeneous network Embedding based approach for Recemendation)**: the HeteCF method is based on a social collaborative filtering algorithm using heterogeneous relations.

**HeteMF (Dual Similarity Regularization)**: the HeteMF method is based on the HIN recommendation method through combining user ratings and item similarity matrices.

**CMF (Dual Similarity Regularization)**: The CMF method is based on the coupled matrix factorization recommendation method integrating user couplings and item couplings into the basic MF model.

*4.2. Parameters Analysis*

In order to determine the best learning rate, we evaluated the MFNN model using learning rates of 0.0001, 0.0005, 0.001, and 0.005. The results are shown in Figure 5. Figure 5 shows that there were fewer differences when the learning rate was 0.0005, 0.001, or 0.005. We further analyzed the HR and NDCG values to select the best rate. According to HR, the performance was better when the learning rate was 0.001 rather than 0.005. It is also clear that the value was higher when the learning rate was 0.001 rather than 0.0005 during early training. The trends of NDCG values were similar to those of HR. Thus, we concluded that a learning rate of 0.001 was best in terms of the experimental results.

We used similar methods to determine that the best batch size was 256. We also considered how the number of hidden layers impacts the recommendation performance, in order to determine whether deeper was actually better. We trained the DNN model with 1, 2, 3, and 4 deep layers. The HR and NDCG values achieved with the different numbers of deep layers are shown in Figure 6. As shown in Figure 6, the DNN performed best when the number of hidden layers was 3. In line with these results, we reached the following conclusion: it is not correct to assume that the greater the number of hidden layers, the better the performance is, or vice versa.

Thus, we selected the most reasonable and best number of hidden layers to implement the MFDNN. In addition, the number of embedding factors for MF affected the recommendation performance. We conducted tests using different numbers of embedding factors in order to find the optimal number; the results are shown in Figure 7. Figure 7 shows that, when the number of embedding factors was 32, the MF performance was the best according to HR and NDCG; thus, we set the number of factors to 32.

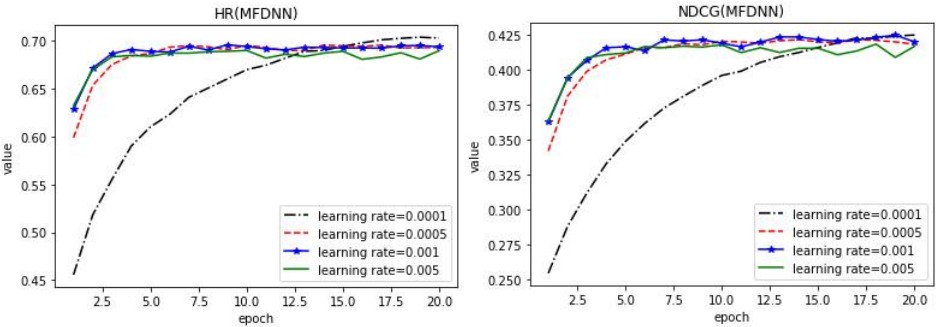

**Figure 5.** The HR and NDCG values for different learning rates.

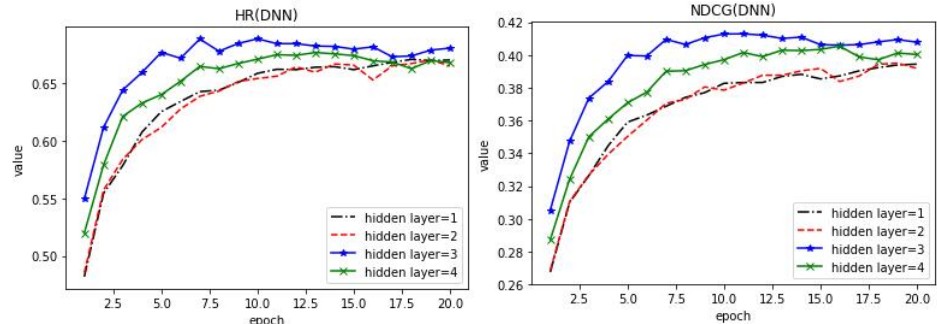

**Figure 6.** The MAP and NDCG values for different numbers of hidden layers.

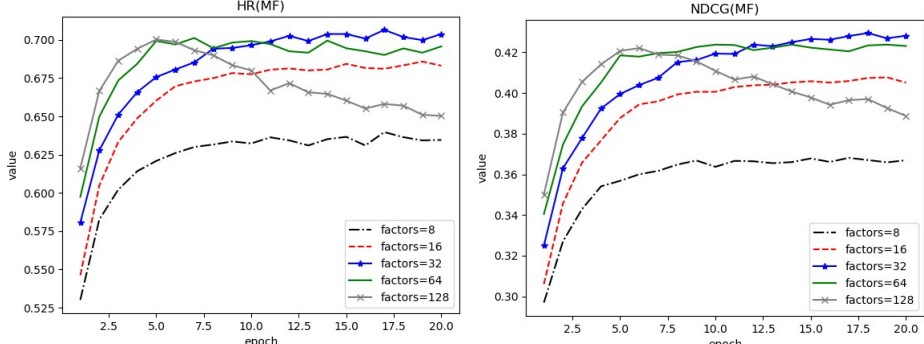

**Figure 7.** The HR and NDCG values for different numbers of embedding factors.

Dropout, a technique for addressing overfitting, refers to the probability that a neuron is kept in the network [72]. We set the dropout rate to be 0.1, 0.2, 0.3, 0.4, 0.5, 0.6, 0.7, 0.8,

and 0.9. As shown in Figure 8, MFDNN was able to achieve its best performance when the dropout rate was 0.6 according to the HR and NDCG metrics. The results illustrate that the robustness of MFDNN was strengthened by adding reasonable randomness.

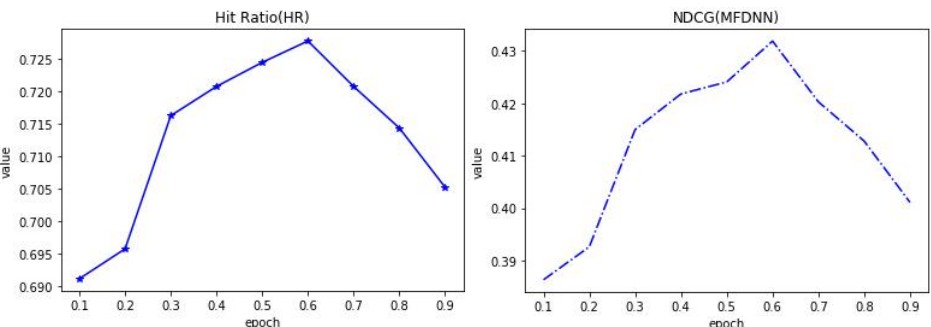

**Figure 8.** The HR and NDCG values for different dropout rates.

### 4.3. Performance and Comparison

The MFDNN recommendation model combines user–item explicit and implicit feedback. The recommendation model combines MF and DNN, which learn embedded features independently; however, the model only merges them in the final output layer through the activation function. In this experiment, we compared and analyzed results in all these aspects. The selected top-N value was taken as N = 10, and the dataset was MovieLens 1m.

(1) Explicit and implicit feedback information

In order to measure the recommendation performance achievable with explicit and implicit feedback information, the explicit and implicit feedback data were separated out for experiments. Specifically, we first removed the implicit feedback information, denoted by MFDNN+, and then we removed the explicit feedback information, denoted by MFDNN-. The experimental results for MFDNN+, MFDNN-, and MFDNN are shown in Figure 9. As shown in Figure 9, the most accurate recommendations were generated using the explicit and implicit feedback data.

According to the HR value, MFDNN provided the highest performance. In the small batch of data before training, the performance of MFDNN- was better than that of MFDNN+, and subsequently the performance of MFDNN+ was better. According to the NDCG value, MFDNN had the best performance, followed by MFDNN- and, finally, MFDNN+. These experiments verified that the combination of explicit and implicit feedback data provided better recommendation performance.

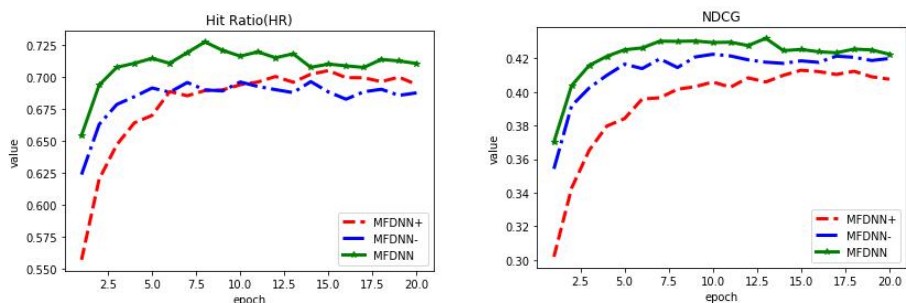

**Figure 9.** The HR and NDCG values achieved with MFDNN+, MFDNN-, and MFDNN.

These results also illustrate the shortcomings of simply considering explicit and implicit feedback data individually. Explicit and implicit feedback data reflect the user's preferences or item feature information from a certain aspect. In order to generate accurate recommendations, it is necessary to fully consider the user's interests or preferences and the potential features of the item itself.

The performance was further verified by considering two bias factors in the explicit feedback information; the results were compared with those from the method that does not consider these two factors (MFDNN–). The experimental results are shown in Figure 10. As shown in Figure 10, when considering two bias factors, the performance of the recommendation model was improved. Although the performance improvement was not very large, the overall performance improved to some extent. Further, in practical applications, different users have different score preferences.

For users who tend to give positive reviews even when they are not satisfied with an item, the score values will not be too low, and the ratings from this type of user are generally high. Conversely, users who are more stringent may give a lower rating even if they are somewhat satisfied with an item; thus, their score values will not be particularly high. In addition, if an item is inexpensive, its overall rating will be higher, and if the item is of poor quality, its overall rating will be lower.

Based on the above analysis, it is meaningful to consider the explicit feedback score preference information, and in theory this can improve the performance of the recommendation algorithm. Although the model considers the bias factor, its performance improvement is not obvious. In subsequent research, it will be necessary to learn more suitable bias factors to improve the recommendation performance [28,73].

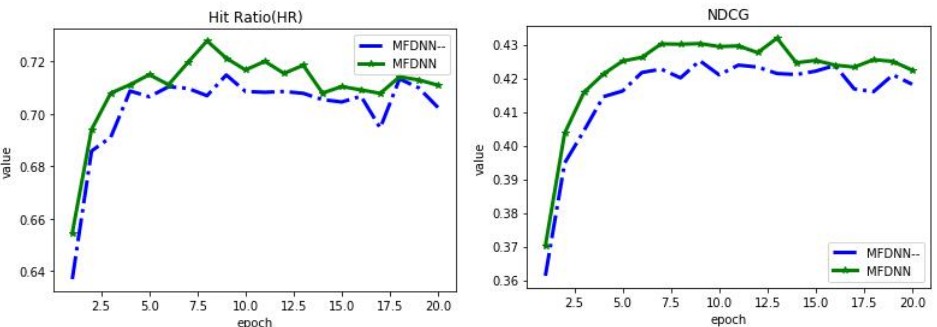

**Figure 10.** HR and NDCG values for MFDNN- - and MFDNN.

(2) Recommendation model based on MF and DNN

MF is a linear model that can mine linear user–item correlation features. On the other hand, DNN is a nonlinear model that can mine the potential nonlinear relationship characteristics of user–item data. In order to verify the recommendation performance of MFDNN, MF and DNN were separately trained as recommendation models. The HR and NDCG values for MF, DNN, and MFDNN are shown in Figure 11. As shown in Figure 11, MFDNN demonstrated the best performance. According to the HR, MFDNN had the best performance, MF and DNN were similar, and their trends were consistent. According to the NDCG value, MFDNN performed best, followed by MF and, finally, DNN. The results of the comparison experiments verify that MFDNN provided the best recommendation performance.

In order to evaluate the recommendation performance of MF and DNN, which learn embedded features independently, separate training processes were employed to facilitate the sharing of the best MF and DNN embedding layers; these are, respectively, denoted as MFDNN: (share MF) and MFDNN: (share DNN). The HR and NDCG values for MFDNN: (share MF), MFDNN: (share DNN), and MFDNN are shown in Figure 12. Figure 12 shows that, although there are small batches of data indicating that the performance was the best when embedding layers were shared, the overall trend shows that MFDNN performed the best.

According to the HR, MFDNN provided the best overall performance, while MFDNN: (share MF) performed better than MFDNN: (share DNN) at the early epochs; subsequently, the performance of MFDNN: (share MF) decreased. According to the NDCG, the overall trend is consistent with HR. Optimal performance was achieved when MF and DNN learned embedding independently.

(3) Comparison with the baseline methods

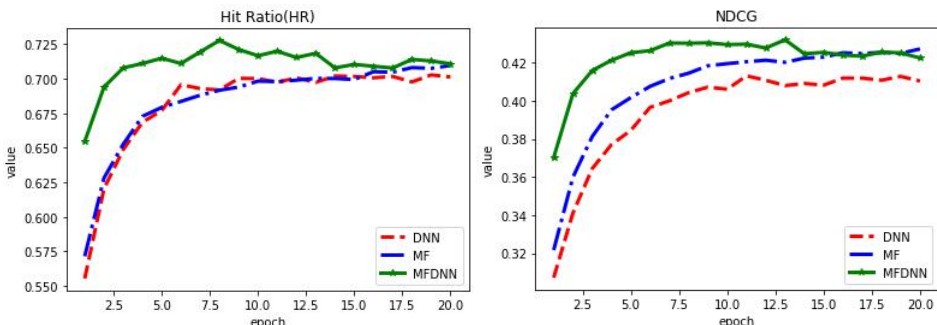

**Figure 11.** HR and NDCG values for MF, DNN, and MFDNN.

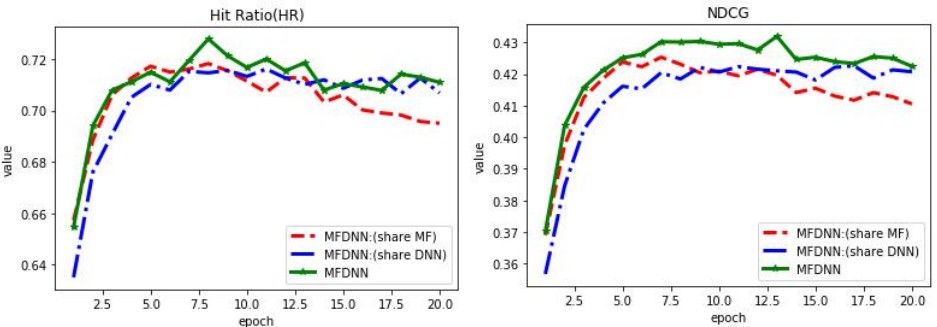

**Figure 12.** The HR and NDCG values for MFDNN: (share MF), MFDNN: (share DNN), and MFDNN.

In order to evaluate the performance of MFDNN over other recommendation models, we trained two state-of-the-art recommendation models (DMF and NCF) and three HIN-based methods (HeteCF, HeteCF, and CMF) separately. The HR and NDCG values for MFDNN and the two baseline methods are shown in Figure 13 (MovieLens 1m) and Figure 14 (Netflix). The HR and NDCG values for MFDNN and the three HIN-based baseline methods are shown in Figure 15 (MovieLens 1m) and Figure 16 (Netflix). The best performance results of each baseline method are shown in Figure 17.

Figures 13–16 show that MFDNN achieved the highest performance. According to the HR, MFDNN performed the best, followed by the two baseline methods NCF, DMF, and the HIN-based baseline methods, HeteCF, HeteMF, and CMF. According to the NDCG value, MFDNN performed the best, followed by the two baseline methods, NCF, DMF, and the HIN-based baseline methods, HeteCF, HeteMF, and CMF. Further analysis indicates that the performance of NCF was better than that of DMF. As NCF uses implicit feedback data and DMF uses explicit feedback data, the results are consistent with the experimental results shown in Figure 9. Table 4 shows the best HR and NDCG values for the MFDNN and baseline methods.

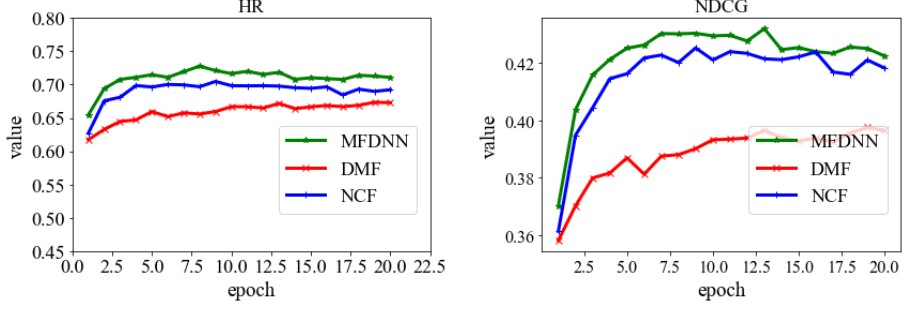

**Figure 13.** The HR and NDCG values for the MFDNN and baseline methods.

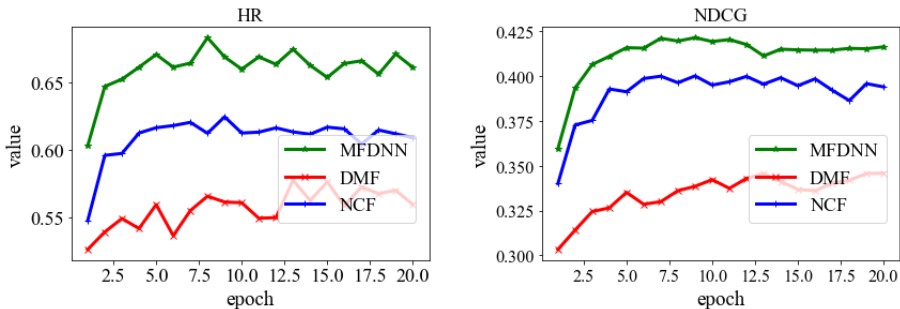

**Figure 14.** HR and NDCG values for MFDNN and baseline methods.

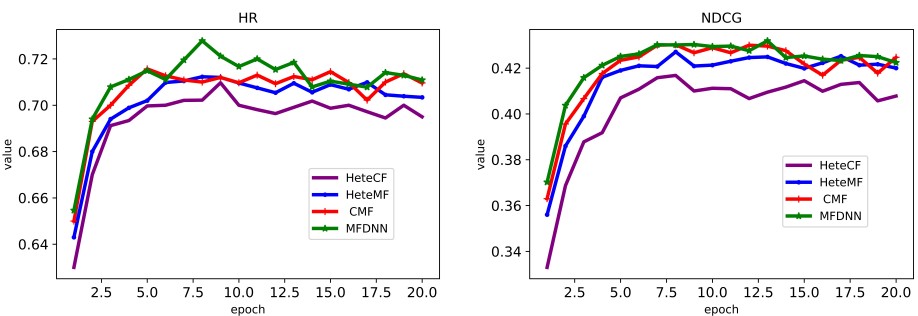

**Figure 15.** The HR and NDCG values for the MFDNN and HIN-based baseline methods .

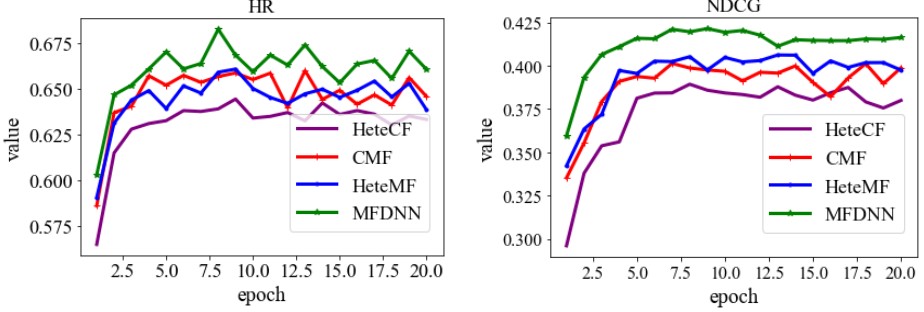

**Figure 16.** The HR and NDCG values for the MFDNN and HIN-based baseline methods .

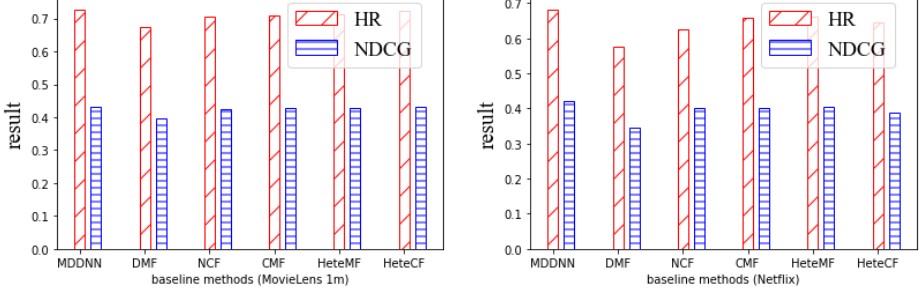

**Figure 17.** The results of the MFDNN and baseline methods.

**Table 4.** The experimental results compared with the results from the baseline methods.

| Method | MovieLens 1m | | Netflix | |
| | HR@10 | NDCG@10 | HR@10 | NDCG@10 |
|---|---|---|---|---|
| MFDNN | **0.7278** | **0.4319** | **0.6828** | **0.4214** |
| DMF | 0.6735 | 0.3975 | 0.5776 | 0.3459 |
| NCF | 0.7048 | 0.4252 | 0.6245 | 0.4000 |
| HeteCF | 0.7097 | 0.4268 | 0.6601 | 0.4013 |
| HeteMF | 0.7123 | 0.4271 | 0.6609 | 0.4062 |
| CMF | 0.7235 | 0.4308 | 0.6445 | 0.3893 |

Table 4 clearly shows that MFDNN outperformed the two baseline methods. In terms of HR, MFDNN achieved an average 5.4% improvement over DMF, an average 2.3% improvement over NCF on the MovieLens 1m datasets. In terms of HR, the two compared methods underperformed MFDNN by an average of 2.75% in terms of HR on the Netflix datasets. The five methods underperformed MFDNN by an average of 3.75% in terms of NDCG. The performance improvements provided by MFDNN are statistically significant according to these results.

(4) Selection of N of Top-N

In the above experiment, the value of N was 10; however, with different N values, the HR and NDCG values will also be different. We selected N values of 5, 10, and 15 to train the model and chose the best performance in the training epochs. The HR and NDCG values for different top-N values are shown in Table 5:

**Table 5.** The HR and NDCG values for different top-N values.

| Top-N | MovieLens 1m | | Netfilx | |
| | HR | NDCG | HR | NDCG |
|---|---|---|---|---|
| N=5 | 0.5303 | 0.3672 | 0.4583 | 0.3058 |
| N=10 | 0.7279 | 0.4319 | 0.6828 | 0.4214 |
| N=15 | 0.7869 | 0.4443 | 0.7542 | 0.4386 |

Table 5 clearly shows that MFDNN performed the best when N was 15; however, we cannot conclude that the larger the value of N, the better the performance. HR relates to whether a test item is in the recommendation list; therefore, for this metric, the larger the value of N, the better the performance. The NDCG relates to the order of the test items in the recommendation list.

*4.4. Discussions*

In this section, we further analyze the architecture of MFDNN and discuss the experimental results to illustrate the performance of MFDNN.

(1) Table 4 clearly shows that MFDNN outperformed the other baseline methods, which indicates that MFDNN improved the top-N recommendation performance of HINs to an extent.

(2) We see that, by combining explicit and implicit feedback information, we can improve the recommendation performance significantly (+3.0% and +2.5%, respectively, for MFDNN+ and MFDNN- in terms of HR, and +1.9% and +1.0%, respectively, for MFDNN+ and MFDNN- in terms of NDCG).

(3) We also found that configuring MF and DNN to learn embedding factors independently improved the recommendation performance (+0.6% and +0.7%, respectively, for MFDNN+ and MFDNN- in terms of HR and +0.8% and +0.9%, respectively, for MFDNN: (share MF) and MFDNN: (share DNN) in terms of NDCG. Although it is not obvious, learning embedding factors independently can improve the performance to a certain extent.

Although MFDNN provided significantly improved performance compared with baseline methods, there is room for further improvement in terms of the metrics. We will continue our best efforts to improve the performance of MFDNN.

## 5. Conclusions

In this work, we explored the information behind the meta-path in the binary network. We designed a new framework MDFNN. The model considers both the explicit feedback information and implicit feedback information of the user-object. It fully captures the preference information of the object based on the meta-path and merges the obtained information into the MFDNN to mine the user–item linear and non-linear characteristics. We proved the rationality and effectiveness of MFDNN through a large number of experiments on various data sets and achieved improvements to existing models. In our comparative experiments, MFDNN was superior to the five models in terms of HR and NDCG.

Although MFDNN improved the recommendation performance, there are still other factors that we should consider. On the one hand, we excavated certain potential features, and there are other available features that have not been excavated, such as other semantic information. On the other hand, we still need to improve the operating efficiency. This takes longer to run on a data set with a large amount of data. This work explores the potential of using explicit and implicit information to mine the potential information of the meta-path in recommendation.

In addition to the meta-path information used in this article, there is other potential information in the real scenes, such as the mining of semantic information of gender, age, and time; another exciting direction is to apply the model to other realities in the scene or use other structural information of the real scene, such as social networks and project context. In social networks, through the combination of social network information, we can also investigate how social influence affects recommendations.

**Author Contributions:** Conceptualization, J.G., Y.S., and X.Z.; Project administration, X.Z. and J.G.; methodology, X.Z.; software, X.Z.; validation, Q.L., C.W. and Z.Z.; formal analysis, Q.L.; investigation, Y.S.; resources, S.W.; data curation, X.Z.; writing—original draft preparation, Y.S.; writing—review and editing, X.Z.; visualization, Z.Z. and S.W.; supervision, Y.S.; All authors have read and agreed to the published version of the manuscript.

**Funding:** This research received no external funding.

**Conflicts of Interest:** The authors declare no conflict of interest.

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
