# Peer review of "A Top-N Movie Recommendation Framework Based on Deep Neural Network with Heterogeneous Modeling"

_applsci, doi:10.3390/app11167418_

Round 1
Reviewer 1 Report
In this submission, the authors propose a recommendation algorithm that tries to fuse explicit and implicit feedback information, using matrix factorization and deep neural networks. My main concern lies within the experimental part of the submission, as they are only considering a single, small recommendation dataset (Movielens-1M), where additionally the implicit and explicit feedback matrices are fully overlapping. In order to better illustrate the merits of their proposed approach, the authors must consider at least one larger dataset, where the said matrices will not fully overlap.
Some additional points:
1. ItemCF and UserCF approaches are way to baseline to be considered and should be removed
2. The Related Work section should be moved right after the Introduction.
Author Response
Please see the attachment. Thank you very much for your advice.

Reviewer 2 Report
The authors of the paper propose MFDNN, a top-N recommendation method for heterogeneous information networks. They considered explicit and implicit feedback information to mine the preferences of users and underlying features of items. They combined matrix factorization and deep neural network to construct the recommendation model. Using the leave-one-out evaluation methodology, they selected 4 recommendation methods as baseline methods. According to the results of the experiment, the proposed architecture significantly outperformed four baseline methods in terms of HR and nDCG.
The paper is informatory and the method is promising although it definitely requires more experiments.
I suggest to work on the Abstract to concentrate on the merits of the research and most important parameters of the results.
In the Introduction Section, I really like the contributions of the paper presented in bullet points. However, please delve more into describing the research gap. Please better describe the links between that gap and the objective of the paper and research question. Please emphasize why the paper is important.
Please consider moving Section 4 Related Work (or some general parts of it) earlier in the paper.
I would also recommend the authors in the Introduction Section and/or Related Work Section to elaborate on the recent research in the wide field of recommendation systems. First of all, please provide a definition of a Recommendation Systems (RS). Please also explain implicit and explicit feedback, their pros and cons, with reference to other papers dealing with that issue, incl. AISC (Advances in Intelligent Systems and Computing) paper titled: Monitoring human website interactions for online stores; and Sensors paper titled Session recommendation via recurrent neural networks over Fisher embedding vectors.
Please show that the RS field is widely studied nowadays, which confirms a great importance of this research topic as well as your interest in it, in particular during the pandemic times, thanks to replacing the need for human recommendations by better and better recommenders. Please refer to most recent studies in the field, incl. Algorithms journal paper titled Towards cognitive recommender systems; LNDECT paper titled Evaluation of varying visual intensity and position of a recommendation in a recommending interface towards reducing habituation and improving sales; Applied Sciences paper titled Horizontal vs. vertical recommendation zones evaluation using behavior tracking; and Applies Sciences paper titled New vector-space embeddings for recommender systems.
Please also elaborate on the Conclusions section. I would recommend to include a summary of the results, confirming the good performance of the presented method, as well as more information on the experiment’s limitations and future research directions.
There are a few English language errors, please work on that, but overall the paper is easy to read.
Author Response

(The authors gave the same response as above.)

Round 2
Reviewer 1 Report
In the revised version of the manuscript, the authors have addressed most of the issued raised in my initial reviews. However, they should address the following points (hence the minor revision) prior to acceptance:
1. Regarding the Netflix dataset, please explicitly explain how the implicit and explicit feedback do not overlap, as it is not evident in your manuscript
2. In the Related Work section, you should also make a reference to random walk-based approaches like the following:
[1] Wang, Z., Liu, H., Du, Y., Wu, Z., & Zhang, X. (2019). Unified Embedding Model over Heterogeneous Information Network for Personalized Recommendation. In IJCAI (pp. 3813-3819).
[2] Jiang, Z., Liu, H., Fu, B., Wu, Z., & Zhang, T. (2018, February). Recommendation in heterogeneous information networks based on generalized random walk model and bayesian personalized ranking. In Proceedings of the Eleventh ACM International Conference on Web Search and Data Mining (pp. 288-296).
[3] Alexandridis, G., Siolas, G., & Stafylopatis, A. (2015). Accuracy versus novelty and diversity in recommender systems: a nonuniform random walk approach. In Recommendation and Search in Social Networks (pp. 41-57). Springer, Cham
[4] Alexandridis, G., Siolas, G., & Stafylopatis, A. (2013, August). A biased random walk recommender based on Rejection Sampling. In Proceedings of the 2013 IEEE/ACM International Conference on Advances in Social Networks Analysis and Mining (pp. 648-652).
